# Methods and Nutritional Interventions to Improve the Nutritional Status of Dialysis Patients in JAPAN—A Narrative Review

**DOI:** 10.3390/nu13051390

**Published:** 2021-04-21

**Authors:** Yoshihiko Kanno, Eiichiro Kanda, Akihiko Kato

**Affiliations:** 1Department of Nephrology, Tokyo Medical University, Shinjuku, Tokyo 160-0023, Japan; 2Medical Science, Kawasaki Medical School, Kurashiki, Okayama 701-0192, Japan; kms.cds.kanda@gmail.com; 3Blood Purification Unit, Hamamatsu University Hospital, Hamamatsu, Shizuoka 431-3192, Japan; a.kato@hama-med.ac.jp

**Keywords:** frailty, sarcopenia, protein energy wasting, hypercatabolism

## Abstract

Patients receiving dialysis therapy often have frailty, protein energy wasting, and sarcopenia. However, medical staff in Japan, except for registered dietitians, do not receive training in nutritional management at school or on the job. Moreover, registered dietitians work separately from patients and medical staff even inside a hospital, and there are many medical institutions that do not have registered dietitians. In such institutions, medical staff are required to manage patients’ nutritional disorders without assistance from a specialist. Recent studies have shown that salt intake should not be restricted under conditions of low nutrition in frail subjects or those undergoing dialysis, and protein consumption should be targeted at 0.9 to 1.2 g/kg/day. The Japanese Society of Dialysis Therapy suggests that the Nutritional Risk Index-Japanese Hemodialysis (NRI-JH) is a useful tool to screen for older patients with malnutrition.

## 1. Introduction

Hemodialysis (HD) therapy, which has been clinically applied in Japan from about 50 years ago, has since undergone various improvements with the use of various new technologies and drugs, and is now one of the world’s leading therapies. HD is now also safely used in older patients and patients with various complications, in whom such treatment was once considered impossible, and it has hence substantially helped to extend their lifespan. At the end of 2018, 339,841 patients in Japan were reported to be receiving some form of dialysis therapy [1]. Previously, patients with end-stage kidney disease (ESKD) died because they were unable to excrete water and potassium from the body. On the other hand, peritoneal dialysis therapy is expected to become more commonly used in the future, as it plays an important role in home care. However, at present, the number of patients using this method is only 9445, because it must be performed by the patients themselves or by family members. However, owing to the small number of patients, there is a lack of data in Japan, and evidence-based criteria for food intake have not been established to date [2]. The number of patients undergoing renal transplantation therapy has increased in recent years. The number is thought to be approximately 10,000 people, but diet therapy for this group of patients has not been discussed sufficiently [3]. In any case, it has become possible for humans to avoid death by ESKD using these three methods. Although various problems remain with each type of treatment, the quality of these treatments provided in Japan is high, and they have greatly benefited ESKD patients.

A characteristic feature of dialysis medicine in Japan is that the proportion of patients undergoing hemodialysis is overwhelmingly high [1]. There may be various reasons for this [4], but patients in Japan generally undergo HD 3 times a week for about 4 h at a time, which was the standard decided 50 years ago. Furthermore, patients receive guidance that eating more than the amount that can be removed by the 12 h a week of HD is life threatening. This was guidance that began in the early years of HD therapy, when patients were generally younger and hence found it more difficult to suppress their food intake owing to their appetite, and when the efficiency of dialysis was lower than at present. Specifically, patients were encouraged to reduce dietary intake if excessive weight gain between dialysis, hyperkalemia, or hyperphosphatemia was observed. However, many dialysis facilities do not have a registered dietitian, and as other medical staff do not study nutrition during their training period, specific advice regarding actual meals that take into account adequate nutrition cannot be provided to the patients. As a result, many doctors will just look at the laboratory test results, and if there is a value that is above the criteria, they just tell the patients “You are eating too much!” as a caution. If patients who lack knowledge or elderly patients receive this caution, they will feel guilty about eating, and reduce their overall food intake. This may result in many patients with insufficient energy intake, because patients should actually increase their energy intake to compensate for the reduced protein intake. In the past, the average age of HD patients was low, and hence the patients’ body reserves had the capacity to compensate for the lack of energy intake, but as the age of starting dialysis is increasing and dialysis patients are generally aging, it has become necessary to introduce new ideas about nutritional management. That is, it is important to reconsider the guidance that assumes that all patients should follow dietary restrictions.

## 2. Assessment of Nutritional Status of HD Patients

In Japan, as a large proportion of HD patients are older patients, there are more patients who have low nourishment who require an increase in their food intake than patients who require dietary restrictions. Then, how can we identify patients with low nourishment during general clinical practice?

In recent years, the words frailty, sarcopenia, and protein-energy wasting (PEW) [5] have been attracting attention. The 2014 edition of the Sarcopenia Diagnostic Algorithm created by the Asian Sarcopenia Working Group stated that it is necessary to measure the grip strength and walking speed of elderly patients to diagnose these conditions [6]. However, it is difficult to routinely perform such measurements during daily practice in dialysis facilities in Japan, and hence it was difficult to use these measurements as indices of a patient’s nutrition status, resulting in a delay in the identification of patients with low nutrition status. In November 2019, the Asian Sarcopenia Frail Society published the Sarcopenia Diagnostic Criteria 2019 [7]. This new standard states a simple algorithm that enables the identification of patients with a low nutrition status by family doctors and in community medical settings without the need for skeletal muscle mass measurement devices. If family doctors measure the muscle strength or physical function of patients, and if either value does not meet the criteria, they are asked to diagnose the patient as having “possible sarcopenia” and to begin nutritional or exercise therapy interventions. If there is a specialized facility nearby, doctors are recommended to introduce the patient to the facility to receive a definite diagnosis. Muscle strength is measured by grip strength, and the cutoff value is less than 28 kg for men and less than 18 kg for women. Physical function is evaluated by performing the 5-times chair stand-up test, and the cutoff value is 12 s or more. In acute-to-chronic-stage medical settings and in clinical research facilities, the diagnosis of sarcopenia is based on grip strength, physical function, and skeletal muscle mass, as stated in the criteria of the 2014 edition. However, as a new measurement method of physical function, in addition to the 5-times chair stand-up test, a simple physical performance battery (Short Physical Performance Battery) consisting of a balance test, a walking test, and a chair stand-up test has been added, which has increased the choice of measurement methods. As a result, it is expected that diagnoses including possible sarcopenia will become easier, and opportunities for intervention will increase. As an international standard for the diagnosis of low nutritional status, the GLIM criteria were announced in 2018. According to this criteria, after judging that a patient is at risk in the screening, if a patient shows signs of either (1) weight loss, (2) a low body mass index (BMI), or (3) low muscle mass, and if the cause is identified as (1) a decrease in dietary intake or digestive function, or (2) inflammation, the patient is diagnosed as having a low nutritional status. However, owing to differences in physique among individuals of various races, specific numerical values to be used as criteria have not been established, even for weight loss, and hence this is not yet a suitable evaluation method.

## 3. The Impact of Serum Albumin in Malnourished HD Patients

On the other hand, serum albumin level is useful as a daily indicator of low nutrition [8] and predictor of mortality [9] in HD patients. Figure 1 shows serum albumin levels and protein intake (normalized protein catabolism rate: nPCR) of Japanese HD patients from a statistical survey of the Japanese Society of Dialysis Therapy (JSDT) [10]. For example, because the simple standard of low nutrition is a serum albumin level of below 3.5 g/dL, most of the older patients meet this criteria, and hence the possibility of low nutrition in patients is high. In addition, the standard recommended protein intake for HD patients in Japan is 0.9 to 1.2 g/kg/day, but few patients achieve this level (Figure 1). In other words, it is clear that older HD patients have a high possibility of low nutrition, and that their protein intake, which is one of the solutions, is also insufficient. Regarding serum albumin level, specialists in Japan have discussed whether it should be measured in a state close to overflow before dialysis or measured in a concentrated state after dialysis to obtain a proper index for evaluation. 

The basis of this discussion is the old idea of deciding on a particular numerical value and providing guidance based on it. However, the correct approach is to follow the changes in albumin levels of the patient over time, and to change the diet accordingly. However, to establish a conclusion to these discussions, we analyzed statistical survey data (*n* = 96,700; men, 61.5%) from the JSDT [11] using the outcome event of 1.5-year mortality. Laboratory data included BMI, serum albumin, creatinine, and blood urea nitrogen (BUN) levels, which are generally measured monthly in dialysis units in Japan. Bootstrap resampling was used to compare the accuracy in predicting mortalities between pre-HD and post-HD levels using area under the receiver operating characteristic curves (AUCs) adjusted for baseline characteristics. A total of 6442 (6.7%) patients died within a year, and 30,965 (32.0%) of the patients died within 5 years. The adjusted AUCs for predicting the 1-year and 5-year mortalities showed that pre-HD albumin, creatinine, and BUN levels, and pre-HD BMI were more accurate than the post-HD levels (*p* < 0.0001 for each). Pre-HD albumin and creatinine levels showed the highest adjusted AUCs for predicting 1-year mortality (0.613 [95% CI: 0.598, 0.629]) and 5-year mortality (0.591 [95% CI; 0.586, 0.595]).

## 4. Methods to Evaluate and Improve Low Nutritional Status

If most older HD patients are likely to be at risk of low nutrition, establishing adequate interventions is crucial. However, at present, professional evaluation methods are very difficult to use for the evaluation of nutritional status. In general medical care, the only way to evaluate a patient is by routine evaluation methods, such as blood tests and weight measurement. In this situation, patients who show, for example, a high serum phosphorus level and a lot of weight gain would be conspicuous. However, it is necessary to look carefully at patients who are not so conspicuous, to identify those with low nourishment, including patients who are thought to have “good self-management”. For example, it is necessary to consider that patients who have unconsciously lost dry weight during a 6-month period are at risk of low nutrition.

The difficulty of intervening in and preventing low nutrition is caused by the difficulty in identifying patients at risk, as a low nutritional state does not immediately cause any particular symptoms. It is difficult to recover after becoming undernourished, and it is hence important to intervene beforehand. For this reason, the Nutrition Risk Index-Japanese Hemodialysis shown in Table 1 was created as a screening tool for low nutrition. This is based on statistical survey data of the JSDT for assessing nutritional risks regarding life prognoses after 1 year (Table 2). BMI, serum albumin levels, serum total cholesterol levels, and serum creatinine levels can be used in daily practice to assess low nutritional risk [12]. This index can be used not only as a screening tool, but also as an explanation tool for patients. To treat patients with low nutrition who also have frailty or sarcopenia, it is necessary to switch ideas from the previous ways of nutritional guidance to identifying inconspicuous cases of low nourishment.

## 5. Methods to Improve Low Nutritional Status

Usually, an increase or decrease in body weight is proportional to the increase or decrease in dietary intake, and thus the increase or decrease in salt intake. In 2019, the Guidelines for the Treatment of Hypertension issued by the Japanese Society of Hypertension set a target salt intake value of less than 6 g/day. Older individuals, people with renal dysfunction, and metabolic syndrome patients are highly salt-sensitive and a reduction in salt intake is often effective, but for individuals who are frail or receiving dialysis therapy, it is desirable to adjust target salt intake appropriately in consideration of their physique, nutritional status, physical activity, etc. [13]. That is, in frail older persons and the chronic dialysis patients, if low nutrition occurs owing to a low salt intake, the amount of salt intake should be increased. It has been shown in Asians that the intake of other nutrients increases with the increase in salt intake [14], and thus adsorption drugs should be used if the problem of hyperkalemia or hyperphosphatemia appears with increased food intake [15]. The most important point in the management of low nutrition is not to give guidance that limits food intake, which should be increased with great care without being caught up in the laboratory data or traditional dietary standards. There are no fatal or urgent side effects associated with increased dietary intake other than hyperkalemia, and this can be improved over time. Moreover, how much protein intake is necessary for HD patients with frailty or sarcopenia remains unknown, although HD patients without frailty or sarcopenia are recommended to take 0.9 to 1.2 g/kg/day of protein, as shown in Table 3. Even though older adults have been found to have a weaker synthetic response to proteins owing to anabolic resistance compared with younger people [16,17,18], HD patients should still take protein to combat frailty and sarcopenia. Although there are few reports from Japan [19,20], both sarcopenia and frailty are more frequently observed in dialysis patients than non-dialysis patients [21,22]. In addition, if non-diabetic HD patients consume enough energy, skeletal muscle mass is not reduced by the currently recommended amount of protein intake [23]. Therefore, it is first necessary to comply with the current standard intake of 0.9 to 1.2 g/kg/day. In addition, because muscle loss occurs not only by protein deficiency but also by energy deficiency [24], if weight loss occurs despite the intake of the currently recommended amount of protein, it is necessary to consider increasing energy and lipid intake and to reconsider the amount of protein intake. Furthermore, to prevent sarcopenia, it is necessary to perform a combination of exercise therapy and dietary intervention. On the other hand, what are the risks of protein intake of more than 1.2 g/kg/day? Although there was no difference in the muscle area of the abdomen and thighs in patients taking more than 1.2 g/kg/day of protein compared with those taking less, it has been reported that the risk of increase in visceral fat and hyperkalemia increases, and the risk of total death may also be higher in such patients [25]. However, it has also been pointed out that an increase in visceral fat in older HD patients with frailty or sarcopenia may not necessarily be a risk of death [26]. In Japan, according to a survey conducted at the end of fiscal year 2015, the protein intake of dialysis patients was substantially lower than the recommended amount of 0.9 to 1.2 g/kg/day [10], and we hence believe it is important to target the protein consumption of HD patients to the present standard of 0.9 to 1.2 g/kg/day.

### 5.1. Meals for Dialysis Patients

Dietary guidance for dialysis patients is provided in accordance with the dietary standards for chronic kidney disease patients proposed in 2014 by the Japanese Society of Nephrology and JSDT [27]. Facilities with registered dietitians can evaluate patient diets according to the nutritional care process and provide correct guidance following this standard. Of course, more accurate assessments of nutritional status using various specialized techniques are also possible [28], and many reports state that individual guidance by registered dietitians is effective in increasing protein intake [29]. On the other hand, many institutions do not have a registered dietitian, and medical professionals who have not studied nutrition are required to provide dietary guidance. However, guidance centered on conventional food restrictions is not necessarily effective, and on the contrary, it has the risk of causing frailty and sarcopenia.

In Japan, the National Institute of Nutrition and Health issues dietary intake standards every 5 years for healthy people to maintain healthy lives [30]. Table 3 compares the standards of healthy individuals and dialysis patients. Dialysis patients have a slightly lower energy setting than healthy people, but they have almost the same settings for protein and salt. Restrictions are necessary only for potassium, and otherwise they should be recommended to eat similar meals as healthy people of the same age range. When a patient’s routine laboratory data changes to an abnormal value while following the above diet, the diet can be adjusted with trial and error to gradually reach a balance. The idea of letting dialysis patients eat the same meals as healthy people is quite the opposite of the idea of dietary restriction that has been the standard in the daily care of HD patients. In the absence of a nutrition specialist, it is also important to manage patients by such a “think as you go” type strategy.

### 5.2. Medical Interventions

The discussion to this point has been about the intake of appropriate meals by patients at their homes. However, for many older dialysis patients in Japan, even though they are provided an ideal menu by a registered dietitian, and this ideal menu is prepared by the family, it is often not possible for the patient to eat a sufficient amount of it owing to old age and a decreased appetite. Therefore, to improve the nutritional status of these patients, supplements may be used to overcome the deficiency of nutrients. Many oral nutritional supplements are available worldwide, and the effective use of these can increase protein intake. Furthermore, oral nutritional support during HD sessions is recommended, as it is generally considered to improve the nutritional status of patients [31], although there are some contradicting reports [32]. The benefits and risks of this type of support varies with the status of the patient, the type of dialysis session, and institute, and hence it is still difficult at present to come to a general conclusion [33].

Parenteral nutrition during HD is another method to increase the nutritional intake of patients with low nutrition. There is more evidence that parenteral nutrition is effective for improving nutritional status in HD patients than oral support. A prospective, multicenter, randomized, open-label, controlled, parallel-group phase IV clinical trial in 107 maintenance hemodialysis patients with PEW was conducted by Marsen et al. [34]. Patients were randomized into 2 groups receiving standardized nutritional counseling with or without intradialytic parenteral nutrition. Prealbumin levels were significantly increased to over baseline levels after 4 weeks of treatment in patients receiving parenteral nutrition compared with the control group. More patients receiving parenteral nutrition therapy achieved an increase in prealbumin level to greater than 30 mg/L after 16 weeks of treatment (48.7% vs. 31.8%). The increase in prealbumin levels as a result of parenteral nutrition therapy was more prominent in patients with moderate malnutrition (SGA score B) compared with patients with severe malnutrition (SGA score C). Unfortunately, we have no clear data at present regarding the effects of parenteral nutrition on HD patients with low nutrition in Japan, and JSDT is currently considering a prospective study to investigate this point.

## 6. Conclusions

Although high-quality dialysis treatments are available in Japan, with the aging of patients, it is necessary to reconsider the current treatment policies. Nutritional management is at the center of this theme, and it is hence necessary to perform research towards the establishment of intervention methods that are suitable for current dialysis patients in Japan.

## Figures and Tables

**Figure 1 nutrients-13-01390-f001:**
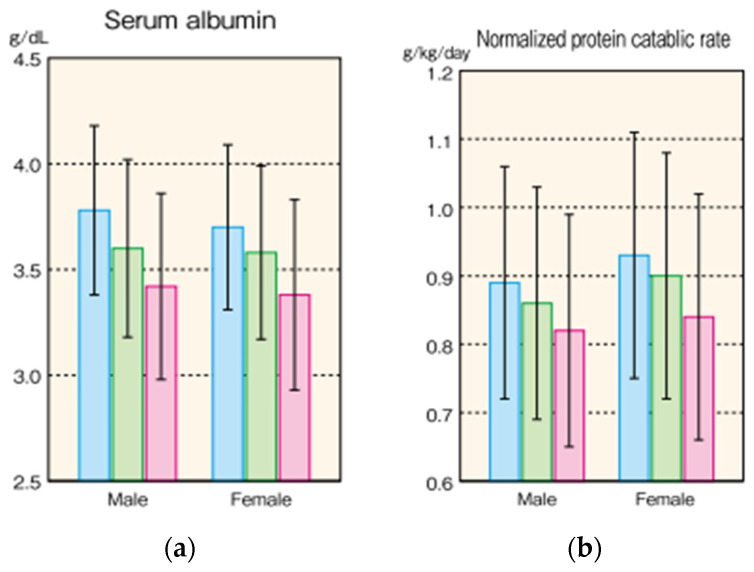
Nutritional status in Japanese HD patients in 2015. (Masakane, I. et al. [10]). (**a**) Serum albumin level (**b**) Normalized protein catabolic rate, of Japanese HD patients. Data are expressed as Mean ± SD. Blue column expresses patients below 60 years old, green column 60–74 years old, and red column over 75 years old.

**Table 1 nutrients-13-01390-t001:** Parameter estimates of the initial sarcopenia model and risk scores (Kanda E., et al. [12]).

	Parameter Estimate	Ratio	Score
Low BMI(≤20 kg/m^2^)	0.51798	3.2555279035	3
Low serum albumin level (age < 65, <3.7g/dL; age ≥ 65, <3.5 g/dL	0.68025	4.275075415	4
Low serum total cholesterol level (<130 mg/dL)	0.15912	1	1
High serum total cholesterol level (≥220 mg/dL)	0.24819	1.559766214	2
Low serum creatinine level (age < 65, male < 11.6 mg/dL, female < 9.7 mg/dL; age ≥ 65, male < 9.7 mg/dL, female < 8.0 mg/dL)	0.65957	4.145110608	4

Each parameter estimate in a Cox proportional hazards model adjusted for baseline characteristics was compared with the smallest parameter estimate (low serum total cholesterol level). Then, the risk scores were determined. Abbreviations: BMI, body mass index.

**Table 2 nutrients-13-01390-t002:** Risk groups and risk of all-cause deaths among HD patients (Kanda E, et al. [12]). Medium-risk and high-risk groups (total score of 8 to 10 and 11 or more, respectively) showed a higher risk of all-cause death than the low-risk group (score: 0 to 7).

	HR	aHR
Low-risk group	Reference	Reference
Medium-risk group	2.94 (2.68, 3.24)	1.96 (1.77, 2.16)
High-risk group	6.99 (6.45, 7.56)	3.91 (3.57, 4.29)

Values are HRs with 98% Cis of medium- and high-risk groups compared with the low-risk group. Abbreviations: aHR, adjusted hazard ratio; CI, confidence interbval.

**Table 3 nutrients-13-01390-t003:** Standard nutrient intake of patients receiving dialysis and healthy subjects in Japan. The standards for HD patients are from the recommendations of JSDT published in Japanese in 2014. The standards for healthy subjects were calculated from the National Institute of Nutrition (Reference 30), using the mean age and mean body weight of Japanese HD patients. Standard body weight is selected and used as body weight. RDA: recommended dietary allowance, DG: tentative dietary goal for preventing lifestyle-associated diseases.

	Energy (kcal/kgBW/day)	Protein (g/kgBW/day)	Salt (g/day)	Potassium (mg/day)	Phosphate (mg/day)	Calcium (mg/day)
Patients (HD 3 times/wk)	30–35	0.9–1.2	<6	≤2000	≤protein (g) × 15	
Healthy men	35–42	1.0	<8	2500 (adequate intake)	1000 (796.5–1062)	700
(66 years old)	(2100–2450 kcal/day)	(RDA 60 g/day)	(DG)	–3000 (DG)	(adequate intake)	(RDA)
Healthy women	34–40	1.0	<7	2000 (adequate intake)	800 (648–864)	650
(68 years old)	(1650–1900 kcal/day)	(RDA 50 g/day)	(DG)	–2600 (DG)	(adequate intake)	(RDA)

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
