# Peer review of "Methods and Nutritional Interventions to Improve the Nutritional Status of Dialysis Patients in JAPAN—A Narrative Review"

_nutrients, 2021, doi:10.3390/nu13051390_

Round 1
Reviewer 1 Report
Monitoring nutritional status of HD patients is important and the authors do a good job of pointing out how the standard of care should be updated to adapt to the improved HD technology and outcomes along with the modern, older patient population.
As it stands, the article is insufficient as a Review paper and does not substantially add to the body of knowledge. There are numerous declarative statements throughout the paper that should be supported by references (e.g. Lines 198,200,260). The English grammar is appropriate throughout the majority of the paper, however, some logical transitions are missing which makes the ideas not flow very clearly at times (e.g. 227-228, 282-283). While the language is mostly appropriate in the paper, the abstract needs considerable revisions.
The Tables and Figures do not appear to be original, but seem to be plagiarized from other published works. If they are original figures, then the captions need to be substantially revised to provide the reader with a clear understanding of what is presented in the Table/Figure. Figure 1 particularly needs a legend as it is not clear what is represented by the colors of the 3 bar graphs.
Author Response
Thank you very much for your comment. We revised our manuscript according to your suggestion.
Monitoring nutritional status of HD patients is important and the authors do a good job of pointing out how the standard of care should be updated to adapt to the improved HD technology and outcomes along with the modern, older patient population.
As it stands, the article is insufficient as a Review paper and does not substantially add to the body of knowledge. There are numerous declarative statements throughout the paper that should be supported by references (e.g. Lines 198,200,260). The English grammar is appropriate throughout the majority of the paper, however, some logical transitions are missing which makes the ideas not flow very clearly at times (e.g. 227-228, 282-283). While the language is mostly appropriate in the paper, the abstract needs considerable revisions.
Thank you very much for your comment. Our statements in lines 198 and 200 are based on reference 11, and also that in line 260 are based on reference 28.
The Tables and Figures do not appear to be original, but seem to be plagiarized from other published works. If they are original figures, then the captions need to be substantially revised to provide the reader with a clear understanding of what is presented in the Table/Figure. Figure 1 particularly needs a legend as it is not clear what is represented by the colors of the 3 bar graphs.
Thank you very much for your comment. The figure and table in this manuscript were cited from reference 8 and 10 with permission. We added a legend for Figure 1.
Reviewer 2 Report
This text is most likely a narrative review, which should be mentioned in the heading for an easier understanding of the article. The text is intrestingly written and summarizes important points in the assessment and treatment of malnutrition in dialysis patients, even if the authors often refer to the results of their own studies. Nevertheless, I would suggest a revision to the paper in terms of structure, abstract and form.
Form:
- Two full stops were set at the end of the abstract.
- In my version, figure 1 as well as table 1A and 1B are displayed in a lower resolution than the rest of the text (maybe screenshots?)
- The heading "Introduction" was written twice in a row
- The numbering of the paragraphs is inconsistent. It begins with "1. Introduction" and is then continued with Roman numerals.
Structure:
- The introduction is easy to understand and eloquently written. However, I think that lines 21-52 are not relevant for the rest of the text and should be deleted or summarized.
- In the second paragraph, "Low nutritional status of HD patients”, many different topics are addressed, which should lead us to the benefits of the NRI-JH as a suitable tool to identify malnutrition in HD patients. I would recommend to name the second paragraph “Assessment of nutritional status of HD patients (line 79-117)
- After presenting the GLIM criteria (line 109-117), I would like a brief explanation of why or whether the authors consider the GLIM criteria to be useful or not for assessing malnutrition in HD patients. Is it possible that the GLIM criteria would classify too many HD patients as malnourished? Perhaps the GLIM criteria would also not be suitable for HD patients, since protein-energy-wasting and protein-energy malnutrition are subject to different influencing factors.
- The third paragraph could be named “The impact of serum albumin in malnourished HD patients” (line 117-163).
- After the sentence “On the other hand, serum albumin level is useful as a daily indicator of low nutrition in HD patients.“ I would like to see some reference to current studies, since older studies did not show consistent results on this topic (e.g. doi: 10.2215 / CJN.10251011) or were only significant in connection with an increased CRP (https://doi.org/10.1371/journal.pone.0190410).
- The fourth paragraph “Methods to evaluate and improve low nutritional status” is the main part of the paper. It introduces the index based on the data from the JSDT. The lines 166-196 contain the “method part”.
- In the next part (“methods to improve low nutritional status”) contains the key messages for the reader regarding alterations in salt- (lines 197-209) and protein intake (209-238) in elderly HD patients.
- The three key messages of the text are:
- The Nutrition Risk Index- Japanese Hemodialysis is a suitable screening tool for assessing malnutrition in older HD patients
- The salt intake in frail or patient should not be restricted if low nutrition occurs
- The protein consumption should be targeted at least at 0.9-1.2 g / kg / day.
- These key points should be made clearer by the authors, as the aim of this article will not be clear to the reader until this point. Neither the title nor the abstract of the article target these key messages. The following title could be suggested “Methods and nutritional interventions to improve the nutritional status of dialysis patients in Japan - a narrative review” The abstract should be rewritten and contain the most important statements of the article. Training medical staff to improve nutritional therapy is still important, but should not be the main content of the abstract.
Author Response
Thank you very much for your comment. We revised our manuscript according to your suggestion.
This text is most likely a narrative review, which should be mentioned in the heading for an easier understanding of the article. The text is interestingly written and summarizes important points in the assessment and treatment of malnutrition in dialysis patients, even if the authors often refer to the results of their own studies. Nevertheless, I would suggest a revision to the paper in terms of structure, abstract and form.
Thank you very much for your kind suggestion. We completely revised abstract with 3 aims you pointed out.
Form:
Two full stops were set at the end of the abstract.
In my version, figure 1 as well as table 1A and 1B are displayed in a lower resolution than the rest of the text (maybe screenshots?)
The heading "Introduction" was written twice in a row
The numbering of the paragraphs is inconsistent. It begins with "1. Introduction" and is then continued with Roman numerals.
We corrected our mistakes in forms.
Structure:
The introduction is easy to understand and eloquently written. However, I think that lines 21-52 are not relevant for the rest of the text and should be deleted or summarized.
In the second paragraph, "Low nutritional status of HD patients”, many different topics are addressed, which should lead us to the benefits of the NRI-JH as a suitable tool to identify malnutrition in HD patients. I would recommend to name the second paragraph “Assessment of nutritional status of HD patients (line 79-117)
After presenting the GLIM criteria (line 109-117), I would like a brief explanation of why or whether the authors consider the GLIM criteria to be useful or not for assessing malnutrition in HD patients. Is it possible that the GLIM criteria would classify too many HD patients as malnourished? Perhaps the GLIM criteria would also not be suitable for HD patients, since protein-energy-wasting and protein-energy malnutrition are subject to different influencing factors.
The third paragraph could be named “The impact of serum albumin in malnourished HD patients” (line 117-163).
After the sentence “On the other hand, serum albumin level is useful as a daily indicator of low nutrition in HD patients.“ I would like to see some reference to current studies, since older studies did not show consistent results on this topic (e.g. doi: 10.2215 / CJN.10251011) or were only significant in connection with an increased CRP (https://doi.org/10.1371/journal.pone.0190410).
The fourth paragraph “Methods to evaluate and improve low nutritional status” is the main part of the paper. It introduces the index based on the data from the JSDT. The lines 166-196 contain the “method part”.
In the next part (“methods to improve low nutritional status”) contains the key messages for the reader regarding alterations in salt- (lines 197-209) and protein intake (209-238) in elderly HD patients.
The three key messages of the text are:
The Nutrition Risk Index- Japanese Hemodialysis is a suitable screening tool for assessing malnutrition in older HD patients
The salt intake in frail or patient should not be restricted if low nutrition occurs
The protein consumption should be targeted at least at 0.9-1.2 g / kg / day.
These key points should be made clearer by the authors, as the aim of this article will not be clear to the reader until this point. Neither the title nor the abstract of the article target these key messages. The following title could be suggested “Methods and nutritional interventions to improve the nutritional status of dialysis patients in Japan - a narrative review” The abstract should be rewritten and contain the most important statements of the article. Training medical staff to improve nutritional therapy is still important, but should not be the main content of the abstract.
Thank you very much for your comment. We revised title, abstract, and structure as your recommendation. We really thank you that our manuscript become much smart.
Round 2
Reviewer 2 Report
Beginning with the introduction, the authors numbered every section. The headings "Meals for dialysis patients" and "Medical interventions" should be subdivided into 5.1. and 5.2. The "Conclusion" should then be labeled 6.
Author Response
Thank you very much for your comment. We re-numbered sections.